# Sustainable Maritime Freight Transportation: Current Status and Future Directions

Suneet Singh [1], Ashish Dwivedi [2] and Saurabh Pratap [1,*]

[1]  Department of Mechanical Engineering, Indian Institute of Technology (BHU), Varanasi 221005, India
[2]  Jindal Global Business School, OP Jindal Global University, Sonipat 131001, India
*   Correspondence: saurabh.mec@iitbhu.ac.in

**Abstract:** Maritime freight has gained popularity among researchers and practitioners due to its cost efficiency and eco-friendly nature. It was initially developed for cargo transfer, but its widespread adoption has made it the backbone of global economy. Despite its favourable nature, some of its serious negative effects have attracted the attention of researchers and scholars. Therefore, the present study reviews the extensive literature available on maritime freight logistics, and evaluates the existing access distance between sustainability practices and maritime freight logistics. A systematic three-stage review process including review planning, review conduct and evaluation is followed in this study. VOSviewer and the R language are used to evaluate relevant issues and changes in the literature. Thereafter, the content analysis highlights the major themes of the subject. This study underscores the impact of innovative technologies discovered to make maritime freight sustainable and also examines maritime freight transport in terms of three pillars of sustainability. The result has implications for policymakers to facilitate the smooth implementation of sustainable practices in maritime freight transportation.

**Keywords:** sustainability; maritime freight transportation; logistics; sustainable logistics; VOSviewer; R Studio

## 1. Introduction

Freight transportation refers to the efficient movement and ideal availability of refined and unrefined components for manufacturers and consumers. With easy access to essentials, it supports production, trade, and consumption activities, and has become a significant part of the supply chain [1]. A developing country such as India carries some 71% of freight by road; nearly 35% of the demand is met by rail, some 6% via waterways, and less than 1% via air [2]. Road freight is the most polluting means of transportation, emanating harmful gasses resulting in external costs [3]. To cover these negatives of road freight transportation, maritime shipping would be a more suitable option [4,5].

Along with this, maritime shipping is considered to be the backbone of global freight trading. According to [6], around 95% of India's international trade by volume and 70% by value is carried out via maritime transport. The compound annual growth of seaborne trade has been 2.9% over the last two decades and is expected to continue growing in the foreseeable future due to the globalisation of manufacturing processes and increased worldwide trade [7]. However, the maritime freight sector is heavily responsible for air pollution. Globally, there are around 96,295 fossil fuels based ships in operation. In the year 2019, these ships accounted for 2.2% of $CO_2$ emissions, 20.98% of $NO_x$, 11.80% of $SO_x$, 8.57% of particulate matter 2.5, and 4.63% of particulate matter 10 [8,9]. In this regard, Monteiro et al. [10] investigated the impact of shipping emissions on air quality in a European context using numerical air quality modelling. The results suggest that shipping emissions partner in the concentration of $NO_2$ in excess of 20% for the deltas, and less than 10% of PM10, in typical urban areas along the west coast located here. Bagoulla and Guillotreau [11] assessed the Greenhouse Gas (GHG) emissions ($SO_2$, $NO_x$, $CO_2$, PM2.5,

PM10) of marine industries over air pollution. Taking into account the ecological impact of maritime freight transport through air pollution, it can be seen that maritime freight transport is among the top five most polluting industries. Maritime transport not only causes environmental issues, but empty container repositioning is also another critical issue for this sector [12].The international trade imbalance among various regions originates from different supply and demand patterns, which accumulate empty containers at demand points. Ultimately, there is a shortage of empty containers to ship cargos to their destination points [13–15]. Subsequently, the reallocation of these empty containers accounts for 30% of all container movement and around 20% of worldwide port operations [16,17]. Lee and Moon [18] propose the idea of foldable containers as an alternative solution to deal with the Empty Container Repositioning (ECR) problem. Toygar et al. [19] identify four main issues along with their sub-issues that arise due to the ECR problem.

To address the environmental and economic issues of sea freight, the concept of sustainability has arisen. Sustainable Maritime Freight Transport (SMFT) is not only a means of providing ecological support, but it also addresses economic and social concerns [20]. In the year 2018, the International Maritime Organization (IMO), the United Nations agency specializing in international shipping, released the preliminary IMO strategy to reduce GHG emissions from ships. This strategy sets a target of 50% reduction in emissions by the year 2050. However, compared to 2008, $CO_2$ emissions from maritime transport are set to climb 90 to 130% by 2050, based on economic growth and energy advancements [21,22]. To meet IMO's expectations, marine industries need to incorporate stringent policy measures to drive down GHG emissions [23]. Indeed, emissions are generated by both ships and ports [24]. Therefore, Gibbs et al. [25] examined how ports can help reduce GHG emissions by developing policies not only for themselves but also for maritime industries. Port managers can create plans or projects to decarbonize the maritime freight sector and also track performance using Emission Inventory (EI) methods [26]. In order to contribute to port sustainability, various researchers have conducted studies on sustainable maritime freight transport. As such, Cammin et al. [27] analyses the barriers to building efficient EIs and highlights some of the key constraints such as poor information systems and data secrecy. Ampah et al. [28] reviewed the literature on cleaner alternative marine fuels and estimated that around 70% of shipping emissions could be saved by the adoption of alternative marine fuels. Similarly, the shipping emissions inventories between river–sea modes and their mixed modes have been compared by [29]. They concluded that river-sea vessels operated in river-sea and mixed mode are more efficient ways of reducing emissions. In this new era of digitization, by adopting technologies such as blockchain technology [30], and big data analytics [31,32], etc., organizations are attempting to be eco-friendly while fulfilling their business requirements [33,34]. In addition, as per the Paris Agreement, the maritime freight industry and its stakeholders are highly encouraged to submit voluntary reports including strengthening and adhering to sustainable development policies as one of their prime objectives [35].

The previously available literature reviews were based on topics such as cleaner alternative maritime fuels [36], digitization [37,38], sustainable maritime supply chains [39,40], sustainable supply chain [41,42], and the circular economy [43,44], etc., in relation to sustainability, but do not provide a holistic picture of the research conducted under the sustainable maritime freight transport paradigm [45,46]. However, a lack of adequate scientific investigation of sustainable maritime development and management may hinder the freight industry's goals for sustainable development, in particular IMO 21's goals on decarbonized maritime freight transport. Thus, through a bibliographic analysis, this paper examines several studies on maritime freight transport.

Bibliographic analysis is a unique and admirable method for discovering and analysing large-scale scientific research. It gives us an insight into the topics thriving in a specific field, and an understanding of the evolutionary nuances for the future [47]. There are many reasons for the adoption of bibliographic analysis such as highlighting the emerging trends

in a specific field and the state of journals, countries and authors' collaboration, and the situation of a particular sector [48,49].

As such, there is still a lack of awareness within the industry and among policy makers. On the other hand, many of the technologies involved in greening industry are still in their infancy. Therefore, the current study serves its purpose by providing a bibliographic overview of the current state of the marine freight industry and the measures being taken to reduce the sector's carbon footprint. At the same time, it will help in building a sufficient basis for taking action in appropriate directions. Subsequently, it will facilitate research by highlighting the gaps and scope of this particular field. In this study, we have developed research questions as follows:

RQ1: What concerns have been raised in the maritime industry regarding sustainability?

RQ2: How can sustainable maritime freight transport combat environmental issues?

RQ3: What are the major contributors to the sustainability of the maritime freight sector?

RQ4: What are the emerging trends and how can the literature in the domain of sustainable maritime freight transport be developed?

The aforementioned research questions will assist in understanding what, how, and where this specific topic has been researched. This document summarizes a review of 378 studies on sustainable maritime freight transport. The study uses the Scopus database, R software and Visualisations of Similarities (VoS) Viewer software.

The rest of the study is organized as follows. A brief introduction to sustainable maritime freight is offered in Section 2. The approach is discussed in Section 3, followed by the results and analysis in Section 4. Section 5 contains content analysis, whereas Section 6 provides discussion. Section 7 concludes the study.

## 2. Overview of Sustainable Maritime Freight Transportation

Sea freight transport is, in a way, an important part of people's daily lives, carrying their necessities such as fuel, food, etc. [50,51]. It is also an important driver of economic competition and jobs [52]. However, in the case of transportation, the main issue is clean energy from an environmental and resource point of view [53,54]. There are other grounds for concern as to the need to cut carbon emissions, which also includes energy conservation [55,56]. The importance of energy conservation lies in meeting our own needs without compromising on the necessities of future generations. To live human life satisfactorily, not only natural but also socio-economic resources are required [57]. SMFT is not only linked to environmentalism. The term "sustainable sea freight transport" also refers to factors such as the economy (efficient and competitive freight transportation), society (inclusive freight transportation), and the environment (green freight transportation) [58,59]. A unified basic framework that defines sustainable maritime freight logistics pictorially and illustrates measures to achieve sustainable sea freight transport is shown (Figure 1):

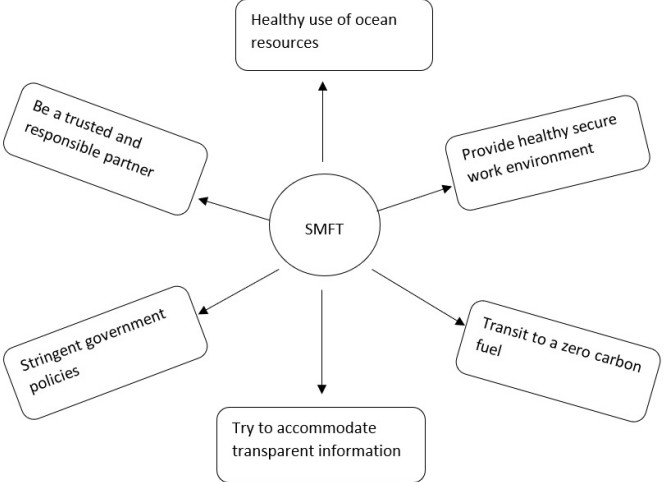

**Figure 1.** Overview of sustainable maritime freight logistics.

## 3. Methodology

This study attempts to highlight the current state of research and future research aspects in the field of sustainability applications in maritime freight, with a particular focus on a systematic review of published records. After that, the study achieves its objectives by reviewing the entire literature systematically.

Through a systematic sequence of literature review, researchers have identified the research gaps by adjusting results in a specific area by assessing, mapping, and analysing the literature [60]. This practice will eventually show the new and correct research direction to future researchers [61]. This study has performed a three-stage systematic literature review suggested by Tranfield et al. [62], which includes the planning phase, operational phase, and outcome review.

In this study, content analysis is used to determine the arguments, methodologies, and applications. The content analysis arrives at its conclusions by extracting useful information from the data originally collected [63].

### 3.1. Planning Phase

The prime objective of this paper is to study and analyse literature relevant to SMFT. As a result, the authors focus on what technologies have been applied and how sustainability has been incorporated into various aspects of maritime freight, or how sustainability has been considered to reduce challenges in terms of $CO_2$ emissions and fuel consumption in ocean freight. We have shortened keywords that refer to sustainability in the field of maritime freight transport. Keywords used in this paper are as follows: "sustainability", "decarbonization", "maritime-freight", "shipping", "sea-borne", and "ocean transportation". We have selected these keywords from previous studies accessed by researchers and through academicians' suggestions.

### 3.2. Conducting the Review and Bibliometric Analysis

Articles from the Scopus database are extracted, using the keywords mentioned in the above paragraph. We only used the Scopus database, as suggested by Nobanee et al. [64]). However, Scopus includes many well-known publications such as the journals of Emerald, Taylor & Francis, IEEE, etc. When we sorted the articles area-wise, there were a total of 1604 submissions, explained in Table 1. Some of them were matched in more than one area, so after sorting them, a total of 811 records were found. Of those 811 studies, 546 were journal articles and 802 English-language articles, the remainder being conference papers or otherwise. In this way, a total of 540 useful articles were extracted from the Scopus database. We have focused our attention on doing a proper review of these 540 records, excluding articles outside our scope. Finally, we considered 540 articles for bibliometric analysis. The search result was stored in a .csv file which included all the article information such as author, author id, abstract, index keywords, etc.

**Table 1.** Search results of articles after refinement.

| Search Keywords | Results (No. of Papers) |
|---|---|
| Environmental science | 359 |
| Energy | 207 |
| Business Management and Accounting | 158 |
| Economics, Econometrics, and Finance | 72 |
| Decision sciences | 55 |
| Engineering | 387 |
| Social sciences | 326 |
| Mathematics | 40 |
| Total | 1604 |
| Excluding Articles matching more than one field | 811 |

Bibliographic analysis has been performed after sorting out the articles. It is useful to researchers in many ways. Through this analysis, we are able to recognize the progress of

intellectual development going on in other countries and their educational institutions and academics. Many scholars have applied this bibliographic analysis in their respective fields: network design of urban freight [65], road freight transportation [66], the energy efficiency of ships [67], advancements made in the ship-port business model [68], and alternative fuels for shipping [69].

Bibliographic analysis consists of steps such as establishing the research objective, material collection, analysis, visual presentation, and elucidation. It provides accurate, chronological, and valuable statistical results as well as illustrating a variety of networks, such as citations and co-citations of journals, authors, bibliographic coupling of countries, universities, and co-occurrence of keywords, etc. The citation network shows how often the study is being cited by others. Various types of co-occurrence networks such as author keywords or keyword networks are also available in the literature. Research articles are represented by nodes in the citation network and their citations by edges. When a third work cites two studies simultaneously, it is called co-citation; as article X and article Y are added to the references of article Z, then article X and article Y are said to be co-cited. Considering the recurrences of the author's keywords or the recurrences of subject keywords in the published works, word frequency calculation is used in bibliometric analysis to determine the study trend in one domain. Consequently, we obtained a list of emerging research trends. Numerous pieces of software, such as CiteSpace, Gephi, BibExcel, etc., are available to accomplish this purpose [70]. In this study, we have applied R Studio and VOSviewer software for qualitative analysis. For more in-depth and better information about the bibliographic analysis, readers can refer to [47]).

*3.3. Material Evaluation*

The aforementioned step includes all the works for co-citation analysis by coding them according to the shortlisted categories. The adoption of sustainability in studies with respect to maritime freight has been classified into various groups to reveal the backbone of the research. Thereafter, all the articles were sorted, and their summaries were integrated, identifying the emergent topics, and recess. Figure 2 depicts a simplified diagram of the research method.

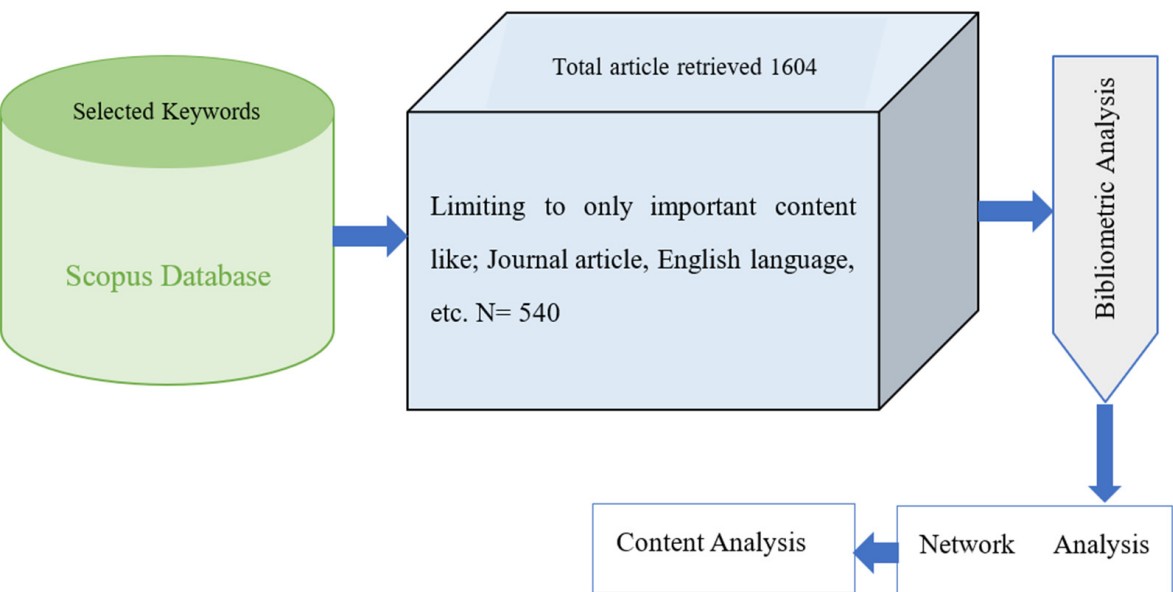

**Figure 2.** Simplified diagram of research methodology.

## 4. Results and Analysis

*4.1. Initial Data Statistics*

From Table 2, it can be observed that researchers are more interested in the lines of our concerned topic recently than a few years back (Figure 3). It can be seen that the total

number of journals published each year almost doubled every 2–3 years. Preliminary numbers reveal that 173 of the 540 papers were published in the top 10 journals, representing approximately 32% of all published papers. The remaining journal statistics are not shown in this table. Table 2 lists those major contributing journals in the field of SMFT, which are as follows: *Sustainability* (Switzerland), *Transportation Research Part D: Transport and Environment*, *Journal of Cleaner Production,* and *Maritime by Holland*.

**Table 2.** Top 10 journals contributing to the area of maritime freight logistics.

| Source | Total | 2013 | 2014 | 2015 | 2016 | 2017 | 2018 | 2019 | 2020 | 2021 | 2022 |
|---|---|---|---|---|---|---|---|---|---|---|---|
| Sustainability (Switzerland) | 62 | 0 | 1 | 1 | 2 | 1 | 5 | 8 | 15 | 18 | 11 |
| Transportation Research Part D: Transport and Environment | 26 | 1 | 1 | 1 | 1 | 3 | 5 | 3 | 3 | 5 | 3 |
| Journal of Cleaner Production | 24 | 0 | 0 | 0 | 1 | 3 | 1 | 1 | 4 | 7 | 7 |
| Maritime by Holland | 12 | 2 | 2 | 2 | 3 | 1 | 2 | 0 | 0 | 0 | 0 |
| Marine Policy | 12 | 0 | 1 | 0 | 0 | 3 | 2 | 0 | 0 | 1 | 3 |
| Maritime Policy and Management | 11 | 0 | 0 | 0 | 0 | 0 | 1 | 1 | 3 | 4 | 2 |
| Transportation Research Part E: Logistics and Transportation Review | 11 | 1 | 1 | 2 | 1 | 1 | 1 | 1 | 2 | 1 | 0 |
| Energies | 10 | 0 | 0 | 0 | 0 | 0 | 0 | 1 | 2 | 6 | 1 |
| Journal of Marine Science and Engineering | 8 | 0 | 0 | 0 | 0 | 0 | 0 | 0 | 1 | 2 | 5 |
| Asian Journal of Shipping and Logistics | 6 | 0 | 0 | 0 | 2 | 1 | 0 | 0 | 1 | 0 | 1 |

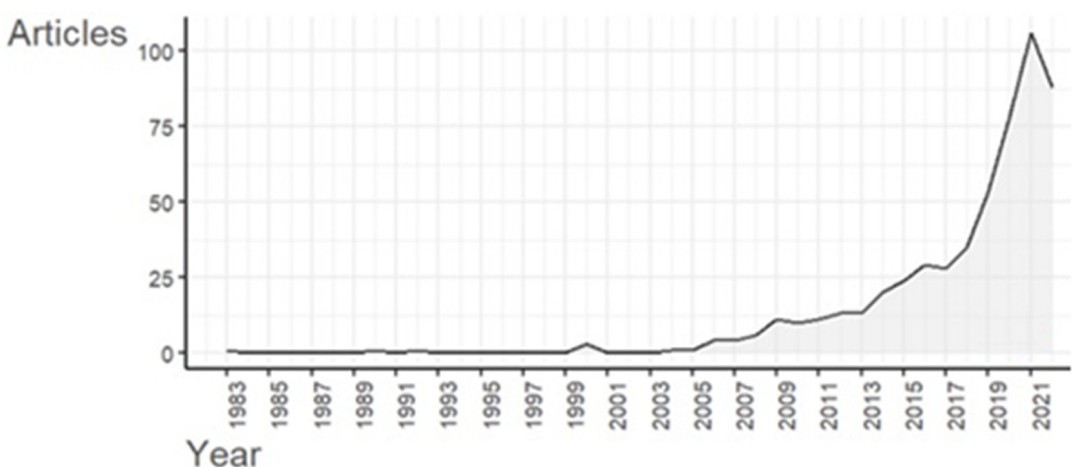

**Figure 3.** Publishing trend in the field of maritime freight logistics.

*4.2. Descriptive Analysis*

In this section, primary information is extracted from the dataset such as the number of keywords, number of articles written per author, number of documents, sources, etc. These data are tabulated so that a detailed analysis of the effective contribution over the years can be made.

Table 3 describes baseline data such as the number of years the work has been studied, the type of studies, etc. To calculate the total number of authors of a manuscript, we need to divide the total number of authors by the total number of works, i.e., 1157/387 = 2.99; whereas, to identify the total co-authors of a manuscript, the average number of authors of a manuscript is considered, i.e., 1366/387 = 3.53. If an author publishes more than one article, it is still counted as one, i.e., the number of co-authors of the article will always be

more than the number of authors per article. The collaboration figure is the proportion of total writers that have conducted their work with multiple authors, and the total studies with multiple authors, which is as follows; $1105/(387 - 52) = 3.29$.

**Table 3.** Basic descriptive statistics based on bibliometric analysis.

| Main Information | | | |
|---|---|---|---|
| **Timespan** | **2017:2022** | **Authors** | |
| Sources | 168 | Authors | 1157 |
| Documents | 387 | Author appearances | 1366 |
| Annual Growth Rate | 25.74 | | |
| Document Average Age | 1.81 | Authors of single-authored documented | 52 |
| Average citations per year per doc | 9.708 | Authors of multi-authored documents | 1105 |
| References | 22,541 | | |
| | | Author Collaborations | |
| Document Type | | Single-authored documents | 52 |
| Articles | 387 | Author per Documents | 2.99 |
| Documents Content | | Co-Authors per document | 3.53 |
| Keywords plus (ID) | 2210 | International co-Authorship% | 30.49 |
| Author's Keywords (DE) | 1440 | Collaboration Index | 3.29 |

*4.3. Statistics Related to Authors, Countries, and Affiliations*

The importance of a researcher in the research field is not just the number of articles attributed to him. By Lotka's rule, we calculated the number of authors who published one article and those who published more than one article in the same time interval, which showed that the number of authors with multiple publications is only a fraction of the authors of a single document. The number of published records from 2017 to 2022 on the topic of Sustainable Shipping based on the Scopus database is 347. These 347 records hold the works of 1000 authors. Out of these 1000 authors, 885 authors have contributed to single manuscripts. Around 30 authors have contributed to the research field with more than three documents. Therefore, it is important to understand the contribution of the parent researchers who have given their valuable time to this field. For this task, the price rule given by Equation (1) is suitable, which gives the lowest number of articles for a researcher.

$$P\_min = 0.749 \times \sqrt{(X\_max)} \tag{1}$$

The above equation defines X max as the number of documents published by a researcher. The authors who played an essential part in this area are Cariou, Lützen, Chang, Imhof, and Kevenaar, computed by Equation (1). Sustainability, supply chain, operations management, maritime logistics, and engineering are the various interest areas of those authors. Their studies mostly revolve around sustainability or carbon reduction techniques.

Information about the countries and institutions involved in the studies is obtained from the original data source. The research into SMFT in various topographic locations is clearly shown in Figure 4. The countries with darker blue colours contributed more to the research writing in this area. The remaining countries also depend on the intensity of the blue colour and data regarding the same is provided in Table 4. We have also collected information about the institutions associated with the researchers. Out of these, the top 10 are displayed in Table 5.

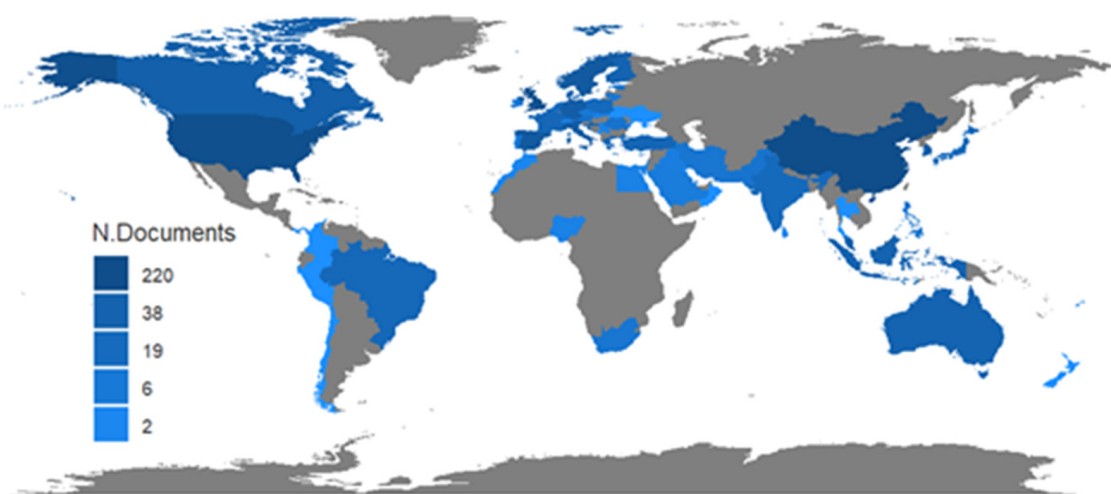

**Figure 4.** The geographical location of all the contributing countries/organizations.

**Table 4.** Top 10 Country-wise statistics (based on the corresponding author).

| Country | Articles | SCP | MCP | Freq | MCP Ratio |
|---|---|---|---|---|---|
| China | 60 | 35 | 25 | 0.111 | 0.417 |
| USA | 43 | 32 | 11 | 0.08 | 0.256 |
| United Kingdom | 40 | 32 | 8 | 0.074 | 0.2 |
| Spain | 22 | 18 | 4 | 0.041 | 0.182 |
| Sweden | 22 | 15 | 7 | 0.041 | 0.318 |
| Singapore | 19 | 7 | 12 | 0.035 | 0.632 |
| Italy | 18 | 12 | 6 | 0.033 | 0.333 |
| Norway | 17 | 13 | 4 | 0.031 | 0.235 |
| Germany | 13 | 8 | 5 | 0.024 | 0.385 |
| Korea | 13 | 8 | 5 | 0.024 | 0.385 |

SCP: Single Country Publications MCP: Multiple Country Publications.

**Table 5.** Top ten universities/institutions statistics.

| Affiliations | Articles |
|---|---|
| Nanyang Technological University | 37 |
| University Of Strathclyde | 36 |
| Dalian Maritime University | 20 |
| University of Malaysia, Terengganu | 20 |
| Shanghai Maritime University | 18 |
| Chalmers University of Technology | 17 |
| Zhejiang University | 16 |
| Technical University of Denmark | 15 |
| Chung-Ang University | 13 |
| Dalhousie University | 12 |

*4.4. Network Analysis*

Publications, authors, and associated institutions are all interconnected. To find out this association, network analysis has been explored. However, it is only a part of the bibliographic analysis. Bibliographic analysis covers networks of co-citations, collaboration networks of authors, and networks of keywords. This analysis captures 347 articles for a visual representation of publications, using VOSviewer software and the R language. All the networks mentioned in this sample are obtained using the VOSviewer software.

4.4.1. Citation Analysis

The citation analysis is performed in the context of local and global citations. Local citations are extracted by computing citations between 387-node networks, and the global

one is responsible for the citations from other studies. An analysis of local citations reflects that 110 articles cited other works in a network of 387 articles. Table 6 displays the top 15 studies with local and global citations.

Yuen [71], writing in *Transportation Research Part E: Logistics and Transportation Review*, has gained the highest number of local citations, although Inees [72] published in *Transportation Research Part D: Transport and Environment* has more global citations than Yuen [71]. This difference in terms of global citations is due to Innes [72] focusing on addressing the environmental sustainability of maritime shipping as well as reducing emissions at ports. The citation network is shown in Figure 5 and the attached results are shown in Table 6. From the below diagram, it can be seen that Yuen [71], Wang [73], Zhou [74], and Altuntas Vural [75] constitute a major group in the citation network.

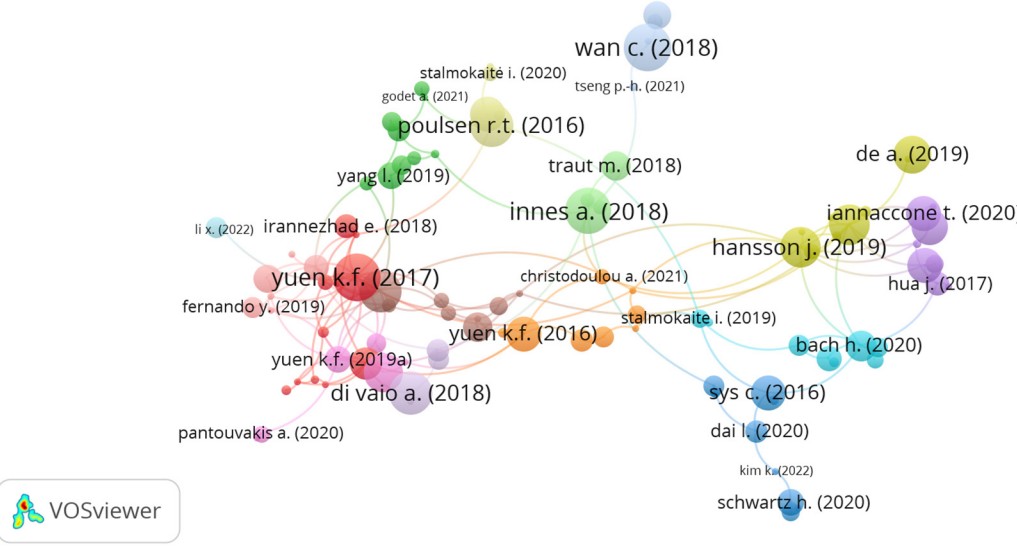

**Figure 5.** Document Citation Analysis.

**Table 6.** Top fifteen cited (Local) documents.

| Document | DOI | Year | Local Citations | Global Citations |
|---|---|---|---|---|
| Yuen et al. [71] | https://doi.org/10.1016/j.tre.2017.10.002 | 2017 | 14 | 73 |
| Hansson et al. [76] | https://doi.org/10.1016/j.biombioe.2019.05.008 | 2019 | 11 | 56 |
| Yuen et al. [77] | https://doi.org/10.1016/j.tre.2018.06.002 | 2018 | 11 | 58 |
| Ren and Liang [78] | https://doi.org/10.1016/j.trd.2017.05.004 | 2017 | 10 | 55 |
| Wang et al. [73] | https://doi.org/10.1016/j.tre.2019.11.002 | 2020 | 8 | 35 |
| Yuen et al. [79] | https://doi.org/10.1016/j.tre.2019.06.014 | 2019 | 7 | 23 |
| Yuen et al. [80] | https://doi.org/10.1016/j.tranpol.2019.03.004 | 2019 | 7 | 22 |
| Stalmokaite and Yliskyla-Peuralahti [81] | https://doi.org/10.3390/su11071916 | 2019 | 6 | 11 |
| Yuen and Lim [82] | https://doi.org/10.1016/j.ajsl.2016.03.006 | 2016 | 6 | 42 |
| Poulsen et al. [83] | https://doi.org/10.1016/j.geoforum.2015.11.018 | 2016 | 6 | 62 |
| Bach et al. [84] | https://doi.org/10.1016/j.trd.2020.102492 | 2020 | 5 | 33 |
| Innes and Monios [72] | https://doi.org/10.1016/j.trd.2018.02.004 | 2018 | 5 | 70 |
| Hua et al. [85] | https://doi.org/10.1016/j.trd.2017.03.015 | 2017 | 5 | 19 |
| Linder [86] | https://doi.org/10.1016/j.trd.2017.07.004 | 2018 | 4 | 17 |
| Sys et al. [87] | https://doi.org/10.1016/j.trd.2015.06.009 | 2016 | 4 | 42 |

### 4.4.2. Co-Citation Analysis

Co-citation maps are used for the visual display of selected data, which is similar to Exploratory Data Analysis (EDA). Various groups can be generated from this analysis, and they can also be delineated based on authors, documents, and journals. Thereafter, co-cited documents, journals, and authors may be revealed and studied. For the present study, a document-based co-cited network is depicted, which is derived from the equation $C = A^T \times A$. In this A stands for a matrix containing a document multiplied by cited

references, and C represents the total co-citations present between documents. A similar measure has been performed to give equal importance to this co-sited network. A better measurement is the power of the relationship deriving from cij/(ci cj) [88]), where cij, ith and jth study denote the recurrence of co-citations between studies, while the notations ci and cj denote the recurrence of independent studies. In order to determine the different groups of articles, data clustering techniques. i.e., the Louvain algorithm, are adopted. This algorithm follows iterative optimization which mainly searches for groups with a high modularity index. Its value lies between [−1, 1]. The modularity index accounts for the compactness of links between groups in contrast with the density of links within groups. For more in-depth information, the detailed discussion can be read in the study by [89]. The co-citations are analysed in R Studio and graphed through VOSviewer. Figure 6 illustrates the co-citation network of articles with at least 20 citations.

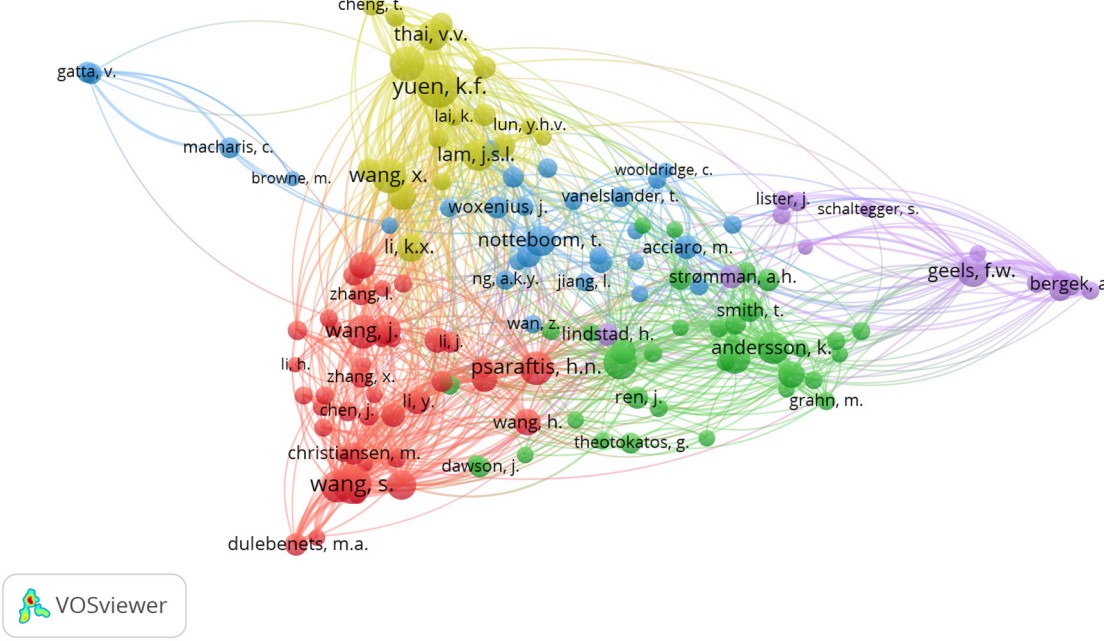

**Figure 6.** Co-citation analysis by cited authors.

### 4.4.3. Keyword Co-Occurrence Network Analysis

Keywords act as a starting point for research. We need these to find any research topic. Accordingly, if any two keywords appear together in the title of the research or the abstract, then it is called co-occurrence. Similarly, a co-occurrence network of keywords is formed according to the simultaneous occurrence of author keywords. Keywords that are very close to each other have a very high co-occurrence rate. This means the length between any two nodes in this graph shows the inverse proportion of the parallelism of those two keywords. Using R Studio and VOSviewer, a total of 467 publications yielded 1592 author keywords, creating a co-occurrence citation graph (Figure 7) from articles with a minimum of five citations.

Figure 8 shows the subject dendrogram. This diagram exposes the hierarchy of keywords and the interrelationships of keywords plus. Boxes and vertical lines in Figure 8 indicate different groups but it does not give a clear and true description of the groups. Nevertheless, the number of clusters can be estimated from this picture. The articles related to sustainable maritime freight transport are mainly divided into four groups.

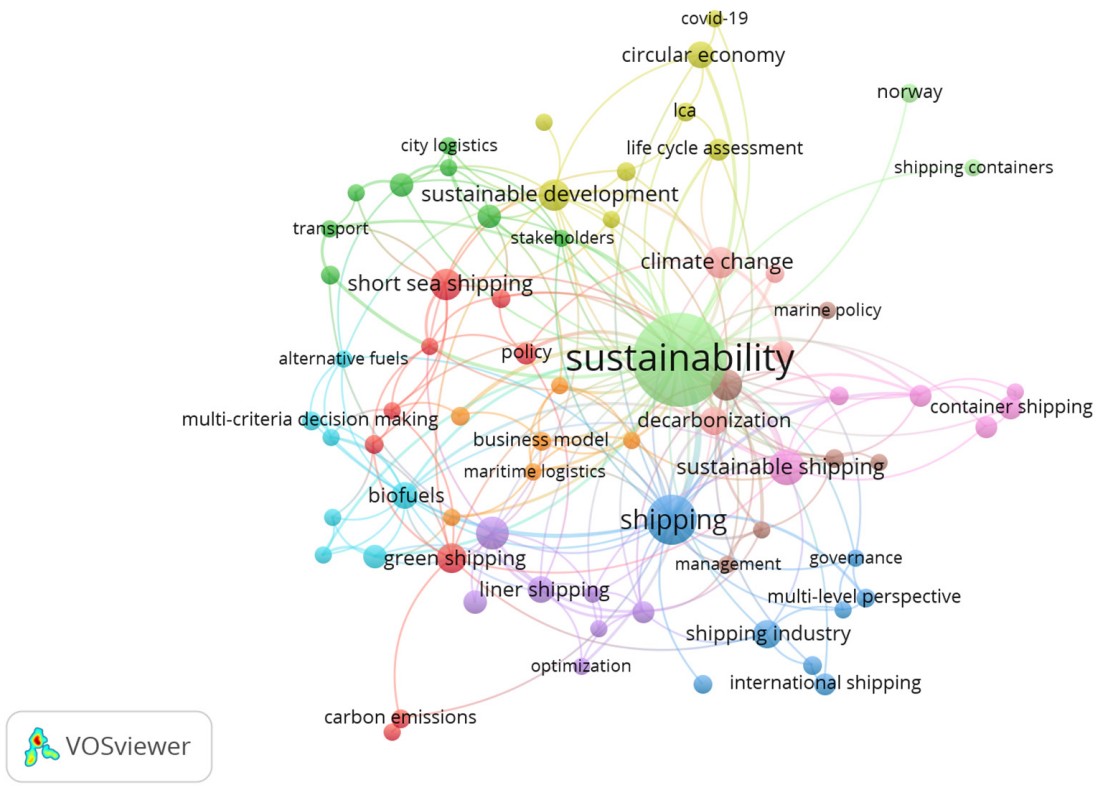

**Figure 7.** Co-occurrence of Author keywords.

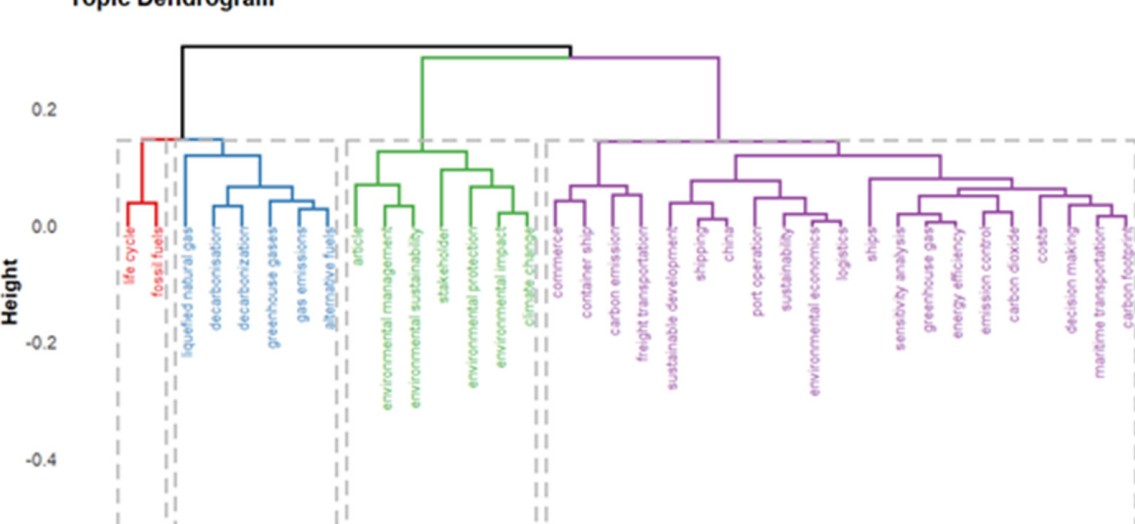

**Figure 8.** Topic dendrogram based on keywords plus.

## 5. Content Analysis

Content analysis on a particular topic is the investigation of research carried out by authors on the same topic. Henceforth, all the articles collected from Scopus are analysed based on the topic, which indicates future research topics for the scholars. In order to capture the above-mentioned data in a well-organized manner, articles are divided based on their role and traits. The studies are divided into six clusters through keywords used in Scopus, illustrated in Figure 9.

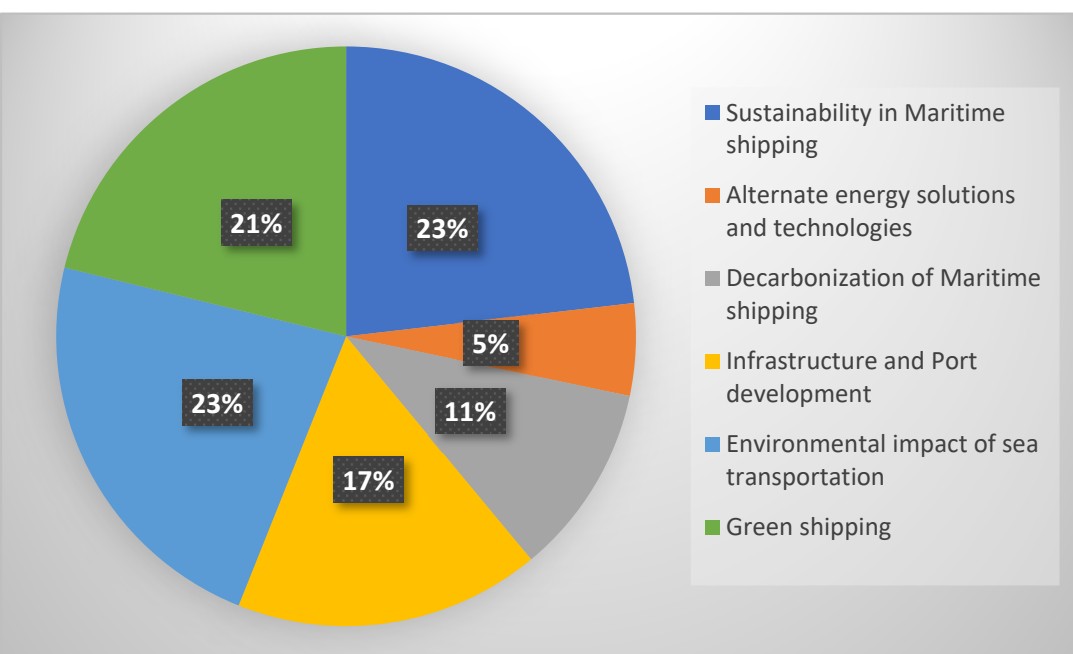

**Figure 9.** Broad classification on the basis of keywords.

### 5.1. Sustainability in Maritime Shipping

Research in marine shipping related to sustainability has contributed 23%, which is the highest; making the shipping sector efficient as well as sustainable. Government and enterprises have also paid attention to sustainability. To make the maritime sector sustainable, blockchain has played a valuable part. Jović et al. [90] explored 20 positive effects of blockchain technology on information trade. Ports play an important role in global trade and, consequently, face social and environmental pressures to provide a sustainable port operation with improved production efficiency and profitability. To achieve this objective, Ashrafi et al. [91] addressed 30 drivers of port sustainability. Pangalos, G. [92] offered a framework to show the myriad problems and disruptions that have affected the shipping industry from a financial standpoint. He also highlighted prospective funding sources for sustainable and renewable energy innovations in the dry bulk shipping sector. Kapidani et al. [93] assessed the level of digital transformation in several developing maritime trade industries, including Montenegro and Serbia. However, a crucial approach to encourage a more sustainable application of sea space is Maritime Spatial Planning (MSP). Accordingly, Frederiksen et al. [94] developed an environmental service-based assessment framework to assess the influence of sustainability in MSP. Rupo et al. [95] proposed a study of the maritime industry with a view to sustainability, open innovation, and value co-creation. Environmental longevity, supply and demand, operations, and port choice are the four main factors that Prajapati et al. [96] identified.

### 5.2. Environmental Impact of Sea Transportation

As we know, the shipping industry is responsible for 90% of total freight transportation. Forecasts of global transport suggest that it will double its current position by 2050 [97]. To meet this growing demand, ships have had to be enlarged, resulting in more problems for maritime enterprises. For example, the number of sea routes is effective from the logistics' point of view but spreads negativity for sea life. Ships are responsible for disturbing the balance of the ocean atmosphere by emitting oil pollution, air pollution, and greenhouse gases. The noise pollution produced by the movement of ships in the sea, the release of ballast water, etc., increases the risk of bio-incursions [98]. The Panama and Suez Canal greatly reduced the emissions of pollutants from shipping by reducing sailing distances. Similarly, Thailand has proposed another canal across the Kra Isthmus to reduce

the fixed distance for ships traveling from East Asia to Europe. Tseng and Pilcher [99] highlight changes in the factors influencing the development of the Kra Canal resulting in negative environmental impacts. They concluded that if ships are made by clean production methods, and technology progresses, then the Kra Canal will prove to be highly beneficial. To achieve the goal of IMO 2050, the first step should be reducing fuel consumption. In this regard, Tillig et al. [97] proposed a clean ship model to reduce the ship's fuel consumption. The study highlights that higher fuel prices would drive lower-speed ships leading to lower fuel consumption as well as limiting $CO_2$ emissions.

### 5.3. Green Shipping

Anthropogenic-sourced shipping contributes 15%, 13%, and 3%, respectively, to annual $NO_x$, $SO_x$, and $CO_2$ emissions [100]. To reduce GHG emissions from marine enterprises, the IMO has laid out a framework for reducing emissions by building ships with the aim of an immediate reduction of 15%, then 20% by 2020, and 30% by 2025 [101]. As a result, currently the shipping industry is going through great stress to follow strict discipline to make its activities clean and green. Therefore, marine enterprises are gradually shedding light on the implementation of green marine practices [102]. These green technologies are very suitable for decarbonization. However, their implementation would require some retrofitting to the ships. In this context, Metzger and Schinas [103] introduced a unique method to determine criteria for adopting green techniques in the shipping industry. Green shipping can be held up as a good example of efficient energy use to move large quantities of goods with low carbon emissions. Many researchers have also made their contributions in this field [104–106].

### 5.4. Infrastructure and Port Development

The globalization of production activities has led to the preponderance of ports in the global supply chain. The function of ports is no longer just handling the loading and unloading of goods; they have become an integral part of the logistics service system at a high level. Shipment reliability and predictability are also important in today's era as the maintenance of products is responsible for a high cost in logistics processes. Despite being so important, the logistics performance and the quality of the infrastructure are often ignored in the literature [107]. Maintaining socio-environmental and economic balance has become a consideration in the construction of all the infrastructure including the port. Taking this thought forward, Lawer [108] described stakeholder participation and environmental and social impact assessment as important weapons to enable port managers to co-create values, avert conflicts, and encourage unified growth. An apparent shift towards a circular economy is the future move, which increasingly reuses or recycles materials and components in the production cycle of products at the end of their life cycle for the benefit of the environment and society. A decision support model for bulk material port operations has been developed [109], which accounts for port dynamics and facilitates improved decision-making across a range of possible circumstances. de Langen et al. [110] provides an overview of the implementation of circular methods in European ports. Other authors have also considered the infrastructure and development of the ports in their research articles [111–113].

### 5.5. Decarbonization of Maritime Shipping

Liquefied natural gas (LNG) helps to decarbonize ocean freight transport, but methane exhaust reduces benefits. Other biofuels and technologies can decarbonize the shipping industry, but they also face many constraints related to their basic costs, required resources, and social acceptance. In addition, making some changes to the ship's design, such as improvements in propeller design, hull design, and cleanliness, can also contribute to decarbonisation by reducing fuel costs [45]. Given that sea freight transport contributes about 3% to GHG emissions, its intensive decarbonization will require financial incentives and policies at the global and regional levels [114,115]). Emissions and suppression measures

fall into two main categories, technical and operational. The International Transport Forum considers alternative fuels, ships' electrification, air support, etc., as other routes to achieve decarbonization [116]. Daniel et al. [117] proposed a roadmap toward greening maritime transport by introducing shore power, as the main option to decarbonize the industry. Psaraftis and Logistics [118] outlined the studies, discussing recent developments, and focusing on the challenges faced by the industry to reduce $CO_2$ adequately. Many other studies in the literature are devoted to decarbonizing the maritime sector and maintaining its sustainability [28,119]. A framework for aligning allocation decisions of berth and ship unloaders in an integrative fashion was proposed by [109]. The eventual objective of these decisions is to cut down on the amount of time spent waiting, working, and dealing with deviations in ship priority.

*5.6. Alternate Energy Solutions and Technologies*

The need of the hour is to use alternative fuels and new technologies in ships to reduce negatives such as GHGs, nitrogen oxides, and sulphur oxides from sea freight transport. Consequently, Bicer and Dincer [120] introduced hydrogen and ammonia as dual fuels to reduce GHG emissions and discovered that transoceanic tankers containing ammonia and hydrogen have significantly less global warming potential than tankers powered by heavy fuel oil. Jeong et al. [121] present a roll-on/roll-off passenger ship using a battery system and highlight its ecological benefits. The results show that the use of battery-powered propulsion reduces global warming capacity by 35.7%, acidification by 77.6%, eutrophication by 87.8%, and photochemical ozone generation by 77%. Hansson et al. [76] assessed seven alternative fuels, for seaborne freight and ranked them concerning four key factors: Social, Economic, Environmental, and Technical. LNG is widely recognized as an alternative fuel in waterborne transportation under the rules of the International Convention for the Prevention of Pollution from Ships. Already, studies have been carried out for two ships in the Taiwan Strait to understand the lifetime emissions of LNG and HFO in which LNG itself was stamped by the board as a future alternative fuel [85]. Xing et al. [122] also conducted a study to find out the most promising alternative fuels for reducing emissions from maritime transport. Table 7 highlights the prominent studies in the field of maritime freight transportation.

**Table 7.** Summary of existing research in the field of maritime freight transportation.

| S. No. | Authors | Title | Results | Method |
|---|---|---|---|---|
| 1 | Dong et al. [123] | Design of a Sustainable Maritime multi-modal distribution network—A case study from automotive logistics | The model proposes a route that makes the most efficient use of both waterways and roadways. | Mixed integer programming |
| 2 | Pérez Lespier et al. [124] | A model for the evaluation of environmental impact indicators for a sustainable maritime transportation system | This study creates an instrument for making decisions in complex environments by quantifying and ranking favoured environmental impact indicators within a multi-criteria decision-making framework (MTS). | Fuzzy AHP and FTOPSIS |
| 3 | Jarašūnienė and Čižiūnienė [125] | Ensuring Sustainable Freight Carriage through Interoperability between Maritime and Rail Transport | The model's viability analysis provided further evidence that the absence of interoperability between maritime and rail transport in international freight carriage is an important issue, and that the developed model could be used to boost the attractiveness of such interoperability. | Theoretical Assessment |

**Table 7.** *Cont.*

| S. No. | Authors | Title | Results | Method |
|---|---|---|---|---|
| 4 | Garg and Kashav [126] | Evaluating value-creating factors in greening the transportation of Global Maritime Supply Chains of containerized freight | Greater Economy of Scale, More Reliability and Predictability, Consolidation, Optimization, and Integration are the top three value-creating factors. | FAHP |
| 5 | Halim et al. [127] | Decarbonization Pathways for International Maritime Transport: A Model-Based Policy Impact Assessment | By utilising all currently available technologies, it may be feasible to achieve nearly complete decarbonization by 2035. Along with this, the price difference between conventional fuels and more environmentally friendly fuel choices must be bridged with financial incentives. | International Transport Forum's International freight model (IFM) and the "ASIF" (Activity, Structure, Intensity, Emission Factor) method |
| 6 | Zis and Psaraftis [119] | Impacts of short-term measures to decarbonize maritime transport on perishable cargoes | In contrast to a speed restriction, a power limit or a goal-based measure would benefit liner shipping companies using more efficient vessels. | An extension of a nested modal split model |
| 7 | Ampah et al. [28] | Reviewing two decades of cleaner alternative marine fuels: Towards IMO's decarbonization of the Maritime transport sector | The USA has been a driving force in the development of this area, which is expanding at a rate of 15.8 percent per year. In terms of alternative transportation fuels, liquefied natural gas (LNG) has received the most attention. | R-studio |
| 8 | Chang and Danao [128] | Green Shipping Practices of Shipping Firms | Industrial institutionalized norms are the most influential element in shipping companies adopting GSP, followed by the firm's environmental strategy and finally, the environmental demand of customers. | Structural Equation Modelling |
| 9 | Tan et al. [129] | Adoption of biofuels for marine shipping decarbonization: A long-term price and scalability assessment | With the help of modern conversion technologies and abundant domestic feedstock in the United States, significant quantities of biofuels can be produced to reach a critical mass and have an effect as alternative marine fuels. | Linear programming model |
| 10 | Prussi et al. [130] | Potential and limiting factors in the use of alternative fuels in the European maritime sector | The paper demonstrates how, even though cost and GHG reduction are key drivers of fuel uptake, other factors such as technological sophistication, safety laws, operator expertise, etc., are not adequately analysed for some solutions, (e.g., ammonia, hydrogen). | Fleets and Fuels" (FF20) modelling |
| 11 | Gore et al. [131] | Cost assessment of alternative fuels for maritime transportation in Ireland | Even though renewable hydrogen is the best choice for meeting future decarbonization goals, it will not be competitive with LNG and methanol until its fuel price is reduced by another 60% or the proposed carbon tax rate is increased by another 275%. | Net Present Value |

**Table 7.** *Cont.*

| S. No. | Authors | Title | Results | Method |
|--------|---------|-------|---------|--------|
| 12 | Moshiul et al. [132] | Alternative Fuel Selection Framework toward Decarbonizing Maritime Deep-Sea Shipping | Technological factors, technology status, expenses, ecological impact, and wellbeing/safety considerations are the most significant criteria for determining the most suitable choices for shipping firm-level players. | Factor analysis and TOPSIS |

## 6. Discussion

The content analysis carried out in the previous section encourages us to identify managerial insights on sustainability implementation in maritime freight transport. It has been observed that efforts are made to implement sustainability in sea freight transport through various measures which help in controlling emissions. The insights gained from the literature review suggest that the benefits of sustainability and the major applications of sustainability in sea freight transport are to increase the efficiency of sea freight transport, with ripple effects, especially in the social, environmental, and economic spheres. Sustainability protects against all risks to human health and the environment, and provides a strategic advantage to maritime freight transport [133].

The present study proposes a bibliographic literature review of overall sustainability in maritime freight transport. Whereas previous studies have analysed the maritime industry in general [134,135], or any one dimension of the marine shipping industry such as the environment [39], in this study the analysis considers all three dimensions of sustainability. The results show that the scope of sustainability research in the literature on maritime freight transport has gradually expanded over the past decades.

Only in the last few years, one can see a sharp increase in publications on maritime sustainability issues. However, more studies appear to be related to the environmental dimension of sustainability. The present study focuses on all three dimensions of sustainability in maritime freight transport. While addressing the three dimensions of sustainability, some researchers have written articles, but those articles are related to a particular area of the maritime industry such as port stability and technical sustainability [136–138]. In general, maritime freight transport has made considerable progress over the years on the issue of sustainability. Nevertheless, some questions remain unanswered, such as how sea freight transport will manage sustainability among its objectives; will these industries really succeed in embracing sustainability, or will they only do so when the value of sustainability outweighs their commercial benefits? For now, the findings suggest that research efforts are on the way to making maritime freight transport more sustainable.

Here, it is necessary to move away from the hope of winning and pay attention to the circumstances, so that the objectives of sustainability can be included along with more benefits to the industries. How will Marine Freight Industries address these mismatched objectives? Can policies and regulations be put in place that encourage industries to adopt sustainability without losing their competitiveness, while maintaining profitability? Undoubtedly, it will take a time-consuming effort until the value of sustainability is equalized in both literature and practice alike.

*Practical Insights and Policy Implications*

The results of this study provide important implications for policymakers; they suggest that the prevailing literature revolves around winning conditions, competence which results from competitiveness, and cost stress. In fact, multi-purpose solutions are increasingly attracting researchers and professionals; as a result, policies must be expanded to take into account all dimensions of sustainability. Policies for maritime freight transport should seek incentives that counterbalance costs, create economic balance, and reduce negatives.

Ocean freight industries tend to be averse to industry policies that threaten their vital interests as witnessed by international industries. Therefore, policies should be designed

with a clear understanding of worldwide sustainability growth. Otherwise, due to increased regulatory pressure, these sea freight companies may start thinking of moving to areas with less regulation. Henceforth, it would be better to think about sustainable freight transport with fewer regulations. However, the current review suggests that ports can be of great help in making freight transport sustainable, as they are in many ways linked to the maritime freight industries. Furthermore, ports cannot escape from the rules of their nation, that is, they will respond to the policies in the shortest possible time.

In fact, sustainability has received intense attention from ocean freight-based research in the last few years. Studies on the interconnection of maritime shipping industries and corporate social development are also expanding, with important implications for policymakers. On the other hand, policies made for Maritime Freight Industries will also need to be clear that they should not affect any other industry, as the Maritime Freight Industry is linked to other industries as well. If something becomes a hindrance for this popular industry with global trade, then other industries related to it may also be affected inadvertently; as well as social, economic, and environmental impacts such as the imposition of a $CO_2$ levy on ships increasing the cost of maritime transport. Therefore, carriers may shift from sea freight transport to road freight transport, which emits comparatively more emissions. Overall, the purpose for which the $CO_2$ levy was introduced has yet to be achieved.

The boom in publications related to sustainability in maritime freight transport reflects a jump in awareness of sustainability. Policymakers should take advantage of this opportunity. This study also provides implications for managers by offering a comprehensive literature review on speed optimization, alternative marine fuels, and efficiency improvements in marine freight industries. In-depth consideration of sustainability, environmental impact, green shipping, infrastructure, port development, decarbonization, $CO_2$ emissions reduction, and alternative energy solutions and technologies may be particularly attractive to practitioners who can apply these concepts to reality. To our knowledge, there is still a lack of an integrated framework that helps practitioners to achieve a holistic approach to sustainability. Many researchers have worked on either one dimension of sustainability or two, but the present work provides a holistic view of sustainability by furthering its discussion in all three dimensions. We hope the proposed study will help practitioners to a great extent. Not only can the growth of research efforts on sustainability frameworks and executions prove to be beneficial for maritime freight transport, but a collaboration with actual practice, and the appropriateness of results, can also be a great help in understanding the complexities of the particular industry. Thus, the quality of the proposed study of marine research, an important sector, can also be enhanced.

### 7. Concluding Remarks

The application of sustainability practices in the field of maritime freight transport has opened many new avenues for making ocean freight transport more efficient. The past few years have attracted many authors to this research area. The present work focuses on how sustainability practices have been incorporated into maritime freight transport. Thereafter, it tries to uncover all possible advantages, drawbacks, challenges, constraints, and opportunities by conducting a well-equipped review of the available literature, and bibliographic analysis. This bibliometric analysis reveals that some specific aspects of maritime freight transport have been extensively studied such as seaborne trade facilitation, environmental impact, alternative fuels, technologies, etc.

The findings of our study demonstrate that research in the area of maritime freight transportation has been on the rise, with scholars from a range of institutions, disciplines, and jurisdictions publishing their findings in prestigious journals with peer review. According to the analysis of the scientific output by nation, research from China appears to have greater influence in terms of citations. However, cost-effectiveness and biofuels are becoming a growing trend in the United States and the United Kingdom, and research activity has increased dramatically in those countries.

According to the identified trends, it can be said that because of increasing pollution, researchers can focus their research on topics such as decarbonization, sustainability, or overall green shipping. Further, research can be carried out on cost analysis, or digitization of maritime freight, with respect to maritime freight and port sustainability.

This manuscript is bound by its limitations. First of all, we have sorted the keywords by linking them to sustainability and maritime transportation. Therefore, the keywords we have picked may not be known across the world. Different keywords can have different clustering; thus, their interpretation will also be different. A subjective classification is based on personal judgment, and this study follows the same subjective classification. Individual judgment cannot be said to be completely reckless but may reveal the number of articles available for overall analysis to be insufficient.

**Author Contributions:** Conceptualization, S.S. and S.P.; methodology, S.S.; software, S.S.; validation, S.S., A.D. and S.P.; formal analysis, S.P.; investigation, S.P.; resources, S.P.; data curation, S.P.; writing—original draft preparation, S.S.; writing—review and editing, S.S.; visualization, S.S.; supervision, S.P. and A.D.; project administration, S.P.; funding acquisition, S.P. and A.D. All authors have read and agreed to the published version of the manuscript.

**Funding:** This research received no external funding.

**Institutional Review Board Statement:** Not applicable.

**Informed Consent Statement:** Not applicable.

**Data Availability Statement:** Not applicable.

**Conflicts of Interest:** The authors declare no conflict of interest.

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
