# Peer review of "Sustainable Maritime Freight Transportation: Current Status and Future Directions"

_sustainability, doi:10.3390/su15086996_

Round 1

Reviewer 1 Report

Dear authors,

congrats to a interesting piece of work. You have spent a lot of efforts in providing a broad overview on the current state and development in maritime freight logistics. The reviewer finds this work valueable and worth to be published in the journal.

The reviewer has only minor issues and little recommendation. On the one-hand, after reading the manuscript it remains some kind of “confusion” how you separate maritime freight LOGISTICS from maritime traffic. Most of the identified trends are related to shipping, to on-board measures, while measures in the field of logistics are seemingly rather low/minor.

In this respect, the reviewer recommends to consider a rephrasing of the title of the manuscript accordingly.

The reviewer recommends to separate the references section into two. One part shall provide sources referenced in the manuscript and another shall contain the remaining articles you covered in your survey.

A spell check is recommended: There are some typos (e.g. missing empty signs after brackets, or figure 8 title Dendrogram – denogram, chapter heading on the last line of a page etc.).

Overall an interesting work, however, the last chapter "7. Concluding remarks" could be improved by giving an outook on how to continue  with the identified trends in this study and what other ways of improvment are possble fromthe uthors point of view after having completed their first review.

The reviewer ishes all the best for your further work and remains with
kind regards

Author Response

PFA

Reviewer 2 Report

The paper presents a very interesting and current topic, which is important for maritime freight transportation. The manuscript analyzed a significant number of representative references from this field. However, before publishing the paper, I suggest a minor revision, primarily regarding technical details.

The introductory section is nicely systematized and written. In the rest of the paper, the used methodology is presented, as well as the results of the conducted analysis.

Perhaps in some chapters, authors could systematize the references and present them in tabular form, including the research focus for each of them. For example, they could be realized such a summary in the Discussion chapter for chapters 5.1 to 5.6 or similar. This is a suggestion for eventually improvement, not a specific request for correction. The authors are under no obligation to accept this suggestion.

I would like to point out a couple of technical things which I noticed while reviewing the manuscript, that need to be changed:

-        In line 19 of the text, delete the space after the word “sustainability”,

-        In line 22 of the text, keep the same form in the text - initial capital letter (Logistics),

-        Keep the same form in the text for term “maritime” (line 14, 35, 37...),

-        Check the sentence (from 208 to 210 line of text),

-        Check the sentence (from 211 to 212 line of text),

-        If Figure 1 is taken from another source, please cite it,

-        In line 313 of the text, delete the space after the word “Analysis”,

-        cij  not define in the formula?

-        In line 389 of the text, insert the space after the number “2050”,

-        Keep the same form in the text for Kra Canal (line 399, 401),

-        In line 411 of the text, insert the space after the number “2025”,

-        Check the sentence (from 415 to 417 line of text),

-        Keep the same form in the text for term “ports” (line 422, 423),

-        In line 486 of the text, insert the space after the word “transport”,

-        Check the sentence (from 496 to 498 line of text),

-        In line 500 of the text, insert the space after the word “sustainability”,

-        Check the sentences (from 536 to 541 line of text),

-        Check the sentences (from 567 to 569 line of text).

Author Response

PFA

Reviewer 3 Report

This paper reviewed the extensive literature on maritime freight logistics. The research has  practical significance and provides implications for policy makers to facilitate smooth implementation of maritime freight sustainability. However, several details need attention. 

1. The research artical is more like a research review. If it is a research paper, the research method and depth are not enough.

2. It is suggested to display the existing research results and methods in a summary table.

3. It is suggested to increase the depth of research and reflect innovation.

It is recommentd to resubmit the manuscript after major revisions.

Author Response

PFA

Round 2

Reviewer 3 Report

Thank you for addressing all the comments for improvement. The manuscript has improved a lot. The type of paper is more like a research review than a research paper. Please further clarify the type of paper.

Author Response

Thanks for the comments. As per the suggestion, we have changed as a research review paper.